# COVID-19 Vaccine Acceptance among ASEAN Countries: Does the Pandemic Severity Really Matter?

**DOI:** 10.3390/vaccines10020222

**Published:** 2022-01-30

**Authors:** An Hoai Duong, Ernoiz Antriyandarti

**Affiliations:** 1Faculty of Accounting, Finance and Economics, Business School, Griffith University, Brisbane, QLD 4111, Australia; 2Study Program of Agribusiness, Faculty of Agriculture, Universitas Sebelas Maret, Surakarta 57126, Indonesia; ernoiz_a@staff.uns.ac.id

**Keywords:** COVID-19 vaccine acceptance, willingness, Coronavirus, pandemic severity, ASEAN

## Abstract

The current study uses data surveyed between August and September 2021 in four ASEAN (Association of South East Asian Nations) countries to identify drivers of COVID-19 vaccine acceptance with different levels of the pandemic severity. It also examines the impact of the drivers on vaccine acceptance. The results show that the number of respondents who accept vaccines significantly dominates that of those who do not. In addition, the number of respondents who get the vaccine if the pandemic becomes more severe dominates that of those if it becomes less severe. Results generated from the logistic regressions show that the impact of the drivers on the COVID-19 vaccine acceptance with different levels of the pandemic severity varies in terms of magnitude and direction. Practical recommendations are made based on the findings.

## 1. Introduction

COVID-19 was declared a global pandemic in March 2020 [1]. However, the first case was detected on 30 December 2019 [2]. Eight months later (the date that the surveys to collect data for the current study started), the pandemic had spread to all continents, and negatively affected every aspect of people’s lives worldwide. For example, it was blamed for a global GDP loss of approximately 6.7 per cent (approximately four trillion USD) in 2020 [3]. More seriously, the virus infected 211,647,672 people and took 4429,460 lives during the period [4]. Several approaches have been applied to stop the pandemic and among them was vaccination. COVID-19 vaccines have been proven to help slow down the spread of the virus and ultimately to stop it by creating herd immunity. Therefore, COVID-19 vaccines are recommended by the World Health Organization (WHO) and many governments [5]. Although the COVID-19 vaccines are believed to help stop the pandemic, many people hesitate to have the vaccine due to several reasons, such as the level of the pandemic severity. In particular, if the pandemic becomes more severe people will rush to be vaccinated and vice versa. The hesitation and associated behaviors prevent us from reaching herd immunity. In addition, this behavior creates an opportunity for more variants such as Delta or Omicron [6] to develop, which may be more difficult to deal with. Vietnam, Indonesia, the Philippines, and Malaysia are among ASEAN countries where the cases and deaths caused by the pandemic were the highest and the vaccine coverage was lowest when the Delta variant of the Coronavirus emerged. The current study endeavors to identify the drivers of COVID-19 vaccine acceptance (“vaccine acceptance” and “willingness to be vaccinated” are interchangeably used in the current study) with different levels of the pandemic severity and to examine the impact of the drivers on the acceptance.

## 2. Research Design, Methodology and Models

### 2.1. Study Design

#### 2.1.1. Selection of Study Locations

Four countries where the cases and deaths caused by the pandemic were highest and the vaccine coverage is lowest in the ASEAN countries were selected. These include Vietnam, Indonesia, the Philippines, and Malaysia.

Vietnam is a good country to be selected for the study for the following reasons: the country has successfully dealt with COVID-19 in the first three waves by applying the 5K (in Vietnamese) rules. The rules are mask-wearing (Khau trang), disinfection (Khu khuan), social distancing (Khoang cach), no gathering (Khong tu tap) and health declaration (Khai bao y te). It can be seen that vaccination was not among the recommended rules [7].

In the fourth wave, which occurred in April 2021 with the emergence of the new variant, Delta (more infectious, more lethal and faster spread speed), the old approaches (quick track of F_0_, quarantine or lockdown, etc.) were not sufficiently applicable and useful anymore. In the meantime, the vaccine campaign had not started. As a result, the outbreak appeared to be out of control. In particular, the number of cases and deaths caused by the pandemic in the nation from 27 April 2021 to 27 August 2021 dramatically increased by 114 and 287 times, respectively. Meanwhile, the total vaccinations per hundred, the number of people vaccinated per hundred and that of people fully vaccinated per hundred estimated on 27 April 2021 was modest, approximately 0.32 [8]. Therefore, vaccination was added to the strategy to cope with the pandemic in Vietnam, as shown in Figure 1.

*Source:* Vietnamese Ministry of Health, 2020.

Indonesia had the highest number of COVID-19 cases in ASEAN during the study period. In particular, the number of cases in the country doubled and the number of deaths tripled between 27 April 2021 and 27 August 2021. Meanwhile, the total vaccinations per hundred, the number of people vaccinated per hundred, and that of people fully vaccinated per hundred estimated on 27 April 2021 were 6.96, 4.35 and 2.61, respectively [8]. In addition, the approaches that the government used to deal with the pandemic, including vaccination, are debatable [9,10].

The number of cases and deaths caused by the pandemic in the Philippines doubled between 27 April 2021 and 27 August 2021. Meanwhile, the total vaccinations per hundred, the number of people vaccinated per hundred and that of people fully vaccinated per hundred estimated on 27 April 2021 were 1.63, 1.41 and 0.22, respectively [8].

Between 27 April 2021 and 27 August 2021 the number of cases and deaths caused by the pandemic in Malaysia increased by four and eleven times, respectively. In contrast, the total vaccinations per hundred, the number of people vaccinated per hundred and that of people fully vaccinated per hundred estimated on 27 April 2021 were 4.21, 2.59, and 1.62, respectively [8]. In addition, Arifin and Musa [11] showed that the number of deaths caused by COVID-19 in Malaysian adults who were unvaccinated significantly dominated that of those who were partially or fully vaccinated. As a result, the lockdown lasted longer than that in the other countries and the negative impact was more severe, especially for children [12].

#### 2.1.2. Sampling

Since the dependent variables are dichotomous, logistic regressions are used. The sample size is calculated based on the following formula, which is adopted from Bujang, Sa’at [13]:n = 100 + x ∗ i(1)
x is an integer representing the event per variable and i is the number of independent variables. The recommended x is 50 and the number of independent variables used in the current study is 48. Therefore, the sample size is 2500. During the study period, vaccines were administered to people aged between 18 and 64, then the vaccine coverage was expanded to other age groups. Between the vaccine shots, there is a waiting period. The waiting time varies from vaccine to vaccine. To save time, other age groups (older and younger people) were vaccinated if vaccines were available.

Google Forms was used to design the questionnaires. The questionnaires were translated into local languages, Filipino, Indonesian, Malaysian, and Vietnamese. Links to the questionnaires were then distributed using email, Facebook, Twitter, Zalo, Viber, and Whatsapp to survey participants. The data in Indonesia were collected between 28 August 2021 and 11 September 2021, those in the Philippines were collected between 27 August 2021 and 16 September 2021, those in Vietnam were collected between 16 August 2021 and 27 August 2021 and those in Malaysia were collected between 16 August 2021 and 14 September 2021.

### 2.2. Research Methodologies, Models and Variable Description

Since the dependent variables are in binary format (1 = willing to be vaccinated against COVID-19, 0 = unwilling to be vaccinated), a binary regression specification is applied [14]. The independent variables are selected based on previous studies, especially those reviewed in the current study.
(2)$$\begin{array}{cc} P(Y_{i}=1|X)=\alpha & +\beta_{1}X_{1i}+\beta_{2}X_{2i}+\beta_{3}X_{3i}+\beta_{4}X_{4i}+\beta_{5}X_{5i}+\beta_{6}X_{6i}+\beta_{7}X_{7i}+\beta_{8}X_{8i}+\beta_{9}X_{9i}+\beta_{10}X_{10i}\\ & +\beta_{11}X_{11i}+\beta_{12}X_{12i}+\beta_{13}X_{13i}+\beta_{14}X_{14i}+\beta_{15}X_{15i}+\beta_{16}X_{16i}+\beta_{17}X_{17i}+\beta_{18}X_{18i}+\beta_{19}X_{19i}\\ & +\beta_{20}X_{20i}+\beta_{21}X_{21i}+\beta_{22}X_{22i}+\beta_{23}X_{23i}+\beta_{24}X_{24i}+\beta_{25}X_{25i}+\beta_{26}X_{26i}+\beta_{27}X_{27i}+\beta_{28}X_{28i}\\ & +\beta_{29}X_{29i}+\beta_{30}X_{30i}+\beta_{31}X_{31i}+\beta_{32}X_{32i}+\beta_{33}X_{33i}+\beta_{34}X_{34i}+\beta_{35}X_{35i}+\beta_{36}X_{36i}+\beta_{37}X_{37i}\\ & +\beta_{38}X_{38i}+\beta_{39}X_{39i}+\beta_{40}X_{40i}+\beta_{41}X_{41i}+\beta_{42}X_{42i}+\beta_{43}X_{43i}+\beta_{44}X_{44i}+\beta_{45}X_{45i}+\beta_{46}X_{46i}\\ & +\beta_{47}X4_{7i}+\beta_{48}X_{48i}+\varepsilon_{i} \end{array}$$

Y*_i_* represents COVID-19 vaccine acceptance (with different levels of the pandemic severity) of the *i*th respondent. In particular, Y*_i_*_1_ represents the COVID-19 vaccine acceptance in which the number of cases in the community increases (1 = willing to be vaccinated), Y*_i_*_2_ represents the COVID-19 vaccine acceptance in which the number of cases in the community decreases (1 = willing to be vaccinated), Y*_i_*_3_ represents the COVID-19 vaccine acceptance in which the number of deaths (caused by the virus) in the community increases (1 = willing to be vaccinated), Y*_i_*_4_ represents the COVID-19 vaccine acceptance in which the number of deaths in the community decreases (1 = willing to be vaccinated) and Y*_i_*_5_ represents the COVID-19 vaccine acceptance in which a new variant of the virus emerges (1 = willing to be vaccinated). It is hypothesized that people will be more willing to be vaccinated if the pandemic becomes more severe, such as the number of cases or deaths in their community increases or a new variant emerges [15].

Independent variables are grouped as follows: The willingness to be vaccinated against COVID-19 in people with different socio-demographic characteristics may not be identical. The socio-demographic characteristics in the current study include age (denoted by X_1_ and measured in years), gender (X_2_, 1 = male), residence (X_3_, 1 = urban), marital status (X_4_, 1 = married or cohabiting), family main income source (X_5_, 1 = yes), savings (X_6_, 1 = yes), gold assets (X_7_, 1 = yes), the number of dependents (X_8_, persons), education (X_9_, 1 = tertiary or above) and employment (X_10_, 1 = employed) [15,16,17,18,19,20,21,22,23,24,25,26,27,28,29,30,31,32,33,34]. In addition, the willingness to be vaccinated against the virus of people with national health insurance (X_11_, 1 = national health insurance member) and private health insurance (X_12_, 1 = private health insurance member) and that of people without insurance is anticipated not to be the same [15,20,23,29]. Access to information (to receive news on the pandemic and vaccines) is believed to help people to have a better and sufficient understanding of the pandemic and vaccines, hence is expected to have an impact on the willingness to be vaccinated. In the current study, the access is represented by the number of information channels to receive news on the pandemic (X_13_, measured in channels) and news on the vaccines (X_14_, measured in channels), the frequency to receive news on the pandemic (X_15_, 1 = daily, 0 = otherwise) and news on the vaccines (X_16_, 1 = daily, 0 = otherwise), the sufficiency of news on the pandemic (X_17_, 1 = sufficient) and on the vaccines (X_18_, 1 = sufficient). Vulnerable people tend to seek protection from medical interventions such as vaccines. In the current study, a person is considered to be vulnerable if he or she belongs to the vaccine priority groups (X_19_, 1 = yes) [35] or lives with someone who belongs to these groups (X_20_, 1 = yes) or has comorbidities (X_21_, 1 = yes) [36] or has been infected with the virus (X_29_, 1 = yes) or has a family member or friend or co-worker who is infected with the virus (X_30_, 1 = yes) [16,18,19,24,28,32]. Similarly, the willingness of people to be vaccinated who are fearful may not be identical to that of those who are fearless. In the current study, it includes the fear of being discriminated against (such as being banned from traveling, entering crowded places such as malls or supermarkets or stadiums, receiving government assistance) if not vaccinated (X_31_, 1 = fearful, 0 = fearless), the fear of catching the virus (X_32_, 1 = fearful, 0 = fearless) and the fear of needles (X_33_, 1 = fearful, 0 = fearless) [15,20,21,26,30,34]. In contrast, people who are concerned about the vaccines and relevant matters tend to hesitate to get vaccinated. In the current study, the respondents were asked if they were concerned about the side effects (X_35_, 1 = concerned), safety (X_38_, 1 = concerned), immunity (X_37_, 1 = concerned), and cost (X_40_, 1 = concerned) of the vaccines. In addition, they were asked if they were concerned about the sufficiency of the official information on the vaccines (X_34_, 1 = concerned), the response procedure if a person has a severe shock after being vaccinated (X_36_, 1 = concerned), the distance between their residence and vaccination site (X_39_, 1 = concerned), waiting time to be vaccinated (X_41_, 1 = concerned), time frame within which to be vaccinated (X_42_, 1 = concerned) and the safety and cleanness of the vaccination waiting room (X_43_, 1 = concerned) [15,21,22,23,30,31,34]. Knowledge can help people to have a better and sufficient understanding of the pandemic and virus. Therefore, the willingness to be vaccinated of people with better and sufficient knowledge in these aspects and that of other people may not be identical [15,19,20,23,28]. In the current study, respondents were asked and tested as to whether they have sufficient knowledge about common symptoms of people infected with the virus (X_22_, 1 = sufficient), on the virus transmission route (X_23_, 1 = sufficient), on the ways to prevent the virus (X_24_, 1 = sufficient), on the spread speed of the virus (X_25_, 1 = sufficient), on the virus fatality (X_26_, 1 = sufficient), on the herd immunity generated by at least 70 per cent of the population having been infected with the virus (X_27_, 1 = sufficient) and generated by there being at least 70 per cent of the population fully vaccinated (X_28_, 1 = sufficient) and on the common symptoms after being vaccinated (X_44_, 1 = sufficient). Four dummy variables are used to represent the nationality of the respondents in the four countries, which are Vietnamese (X_45_, 1 = Vietnamese), Indonesian (X_46_, 1 = Indonesian), Filipino (X_47_, 1 = Filipino) and Malaysian (X_48_, 1 = Malaysian). These factors can be briefly illustrated in Figure 2.

## 3. Results and Discussion

### 3.1. COVID-19 Acceptance and the Pandemic Severity

The respondents were asked if they would be willing to be vaccinated if the number of cases or deaths (caused by the virus) in their community increases or if a new variant of Coronavirus emerges (more severe) and decreases (less severe) or no variants of the virus. Overall, the percentage of respondents who are willing to be vaccinated significantly dominates that of those who are not willing. In addition, if the pandemic becomes more severe the percentage of respondents who are willing to be vaccinated is approximately two per cent higher than that if it becomes less severe.

Details of the number of respondents who are willing to be vaccinated with different levels of the pandemic severity are presented in Table 1 and Figure 3. Figure 3 shows that the COVID-19 vaccine acceptance rate rages between 77 to 80 per cent, depending on the level of the pandemic severity. In particular, the number of respondents (of all ages) who are willing to be vaccinated if the pandemic becomes more severe slightly dominates that of those if it becomes less severe.

People with more work or family responsibilities tend to seek protective medical intervention, such as vaccines, than other people. As expected, the number of respondents with more family responsibilities such as females (as Asian culture), married or cohabited, the family main income source, with dependents or employed who are willing to be vaccinated significantly dominates that of those with fewer family responsibilities. In addition, the number of respondents (in these groups) who are willing to be vaccinated when the pandemic becomes more severe slightly dominates that of those when it becomes less severe.

The number of rural respondents who are willing to be vaccinated significantly dominates that of urban respondents. In addition, the number of respondents in this group who are willing to be vaccinated when the pandemic becomes more severe slightly dominates that of those when it becomes less serious.

The number of respondents with protection such as having health insurance (national or private) or savings or gold assets who are willing to be vaccinated significantly dominates that of those without the insurance or savings or gold assets. Additionally, the number of respondents (in these groups) who are willing to be vaccinated if the pandemic becomes more severe slightly dominates that of those when it becomes less severe.

The access to information and information sufficiency on the pandemic and vaccines can help people better understand the danger or fatality from the virus and the advantages of the vaccines, hence it may have an impact on their willingness to be vaccinated. The access to information in the current study is represented by the number of channels (12). In addition, it is represented by the frequency (daily or not) that respondents receive the information. It is also represented by the information sufficiency (sufficient or insufficient) that they receive.

As expected, the number of respondents with better information access (such as having more channels—more than 6 out of 12 channels to receive news on the pandemic and vaccines—more frequently receiving the information and receiving sufficient information) who are willing to be vaccinated significantly dominates that of those with poorer information access or insufficient information. Additionally, the number of respondents (in these groups) who are willing to be vaccinated if the pandemic becomes more severe slightly dominates that of those if it becomes less severe.

Vulnerable people tend to seek protection more than those who are not at risk. Vulnerable people in the current study include those who belong to the vaccine priority groups, who live with someone belonging to these groups, or who have a family member or friend or co-worker infected with the virus. The results show that the number of these vulnerable respondents who are willing to be vaccinated significantly dominates that of those who are not. In addition, respondents who have comorbidities and those who have been infected with the virus are listed as vulnerable people. The results show that the number of respondents who do not have any comorbidities or have never been infected with the virus and are willing to be vaccinated slightly dominates that of those who have a medical history or have been infected with the virus. Additionally, when the pandemic becomes more severe the number of vulnerable respondents who are willing to be vaccinated slightly dominates that of those when it becomes less severe.

As previously addressed, possessing better or sufficient knowledge can help people better understand the consequences of being infected with the virus and the advantages of being protected by the vaccines. In the current study, knowledge is represented by general knowledge (education) and specific knowledge. As anticipated, the number of respondents with a higher education level and sufficient knowledge (such as the symptoms of people infected with the Coronavirus, the virus transmission route, the ways to prevent the virus, the virus spread speed, the virus fatality, herd immunity, and the common symptoms after being vaccinated) who are willing to be vaccinated significantly dominates that of those with lower education or insufficient knowledge. In addition, the number of respondents (in these groups) who are willing to be vaccinated if the pandemic becomes more severe slightly dominates that of those if it becomes less severe.

Fear forces people to seek protection. In the current study, it includes the fear of catching the virus, being discriminated against if not vaccinated, and needles. As anticipated, the number of respondents with the fear of the above things (including needles) who are willing to be vaccinated significantly dominates that of those who are fearless. Despite the fear, the number of respondents who are willing to be vaccinated modestly increases if the pandemic becomes more severe.

In contrast to fear, if people are concerned about something, they tend to be reluctant to proceed. The current study asks if the respondents are concerned about the information sufficiency of the vaccines, on the side effects of the vaccines, on the vaccine immunity, on the vaccine safety, on the distance between their residence and the vaccination site, on the vaccine cost, on the waiting time to be vaccinated, on the vaccination time frame and on the safety and cleanness of the vaccination waiting room. The results show that the number of respondents who are not concerned about the above causes and willing to be vaccinated significantly dominates that of those without the concern. Despite the concern, the number of respondents who are willing to be vaccinated slightly increases if the pandemic becomes more severe.

The number of Vietnamese, Indonesian and Filipino respondents who are willing to be vaccinated significantly dominates that of those who are not. In addition, the vaccine acceptance of respondents according to the pandemic severity is identical.

### 3.2. The Association between COVID-19 Vaccine Acceptance and the Influential Factors

The impact of influential factors on the COVID-19 vaccine acceptance of the respondents is examined using binary regression. Multicollinearity tests have been conducted and the mean VIF (variance inflation factor) is 2.19, which is significantly lower than ten. In addition, the matrix of correlation shows no correlation coefficients greater than 5.5. These test results indicate that no serious issues of multicollinearity exist [38,39,40,41,42,43,44]. The regression results are presented in Table 2.

As expected, older respondents (X_1_), who usually possess sufficient knowledge, are more likely to be vaccinated. The impact is significant at the one and five per cent level, depending on the level of the pandemic severity. This finding is almost in accordance with that found by Edwards and Biddle [17], Freeman and Loe [18], and Kessels and Luyten [16], but different from that explored by Al-Mistarehi and Kheirallah [24], and Dodd and Cvejic [23]. The literature also shows that elderly people with a medical history tend to hesitate to be vaccinated. However, their hesitation has consequences. For example, during the fourth wave, approximately 50 per cent of deaths (caused by the virus) in Ho Chi Minh City were elderly people with comorbidities and unvaccinated. To deal with this challenge, medical teams are sent to vaccinate these people at home [45].

As anticipated, urban residents (X_3_) are more likely to be vaccinated than their counterparts residing in rural areas. The impact is significant at the one and five per cent level, depending on the pandemic severity. This finding is almost similar to that found by Abedin and Islam [22], and Yoda and Katsuyama [21]. In addition to the fixed vaccine delivery points, mobile vaccination should also be arranged to vaccinate people residing in rural areas, especially in remote areas [46].

Gold assets (X_7_) are considered and used as savings in Asian culture. The results show that respondents with gold assets are less likely to be vaccinated when the number of cases in the community decreases. The impact is significant at the five per cent level.

Respondents with responsibilities such as having a dependent (X_8_) tend to follow precautionary procedures. The results show this behavior. In particular, respondents with a dependent are more likely to be vaccinated if the number of deaths caused by the virus increases in their community. The impact is significant at the ten per cent level. This finding is almost in accordance with that found by Chew, Cheong [29], Freeman, Loe [18], Khubchandani, Sharma [26], Kourlaba, Kourkouni [28] and Liu and Li [25].

Respondents with the national health insurance membership (X_11_) are less likely to be vaccinated. The impact is significant at the one and five per cent level, depending on the pandemic severity. If only clients who are willing to be vaccinated can claim insurance, insured clients may change their vaccine acceptance. Dodd, Cvejic [23] found similar behavior in Australia. The vaccines are still new and being a member of the national health insurance may be the driver of their vaccine hesitancy.

Respondents with more information channels (X_14_, more than 6 out of 12) to receive news on the vaccines are more likely to be vaccinated if the pandemic becomes more severe. The impact is significant at the one, five and ten per cent level, depending on the pandemic severity. Vietnamese are daily updated with news on the pandemic and vaccines through Television and radio programs such as “To Safely Co-exist with COVID-19”, and these channels appear to be efficient [47]. Respondents who daily update with news on the pandemic (X_15_) are more likely to be vaccinated when the pandemic becomes more severe. The impact is significant at the ten per cent level if the number of cases or deaths increases and at the one per cent level if a new variant emerges. Respondents with sufficient information on the vaccines (X_17_) are more likely to be vaccinated when the number of cases or deaths increases. The impact is significant at the one and ten per cent level, depending on the pandemic severity.

As anticipated, vulnerable respondents (X_19_, belong to the vaccine priority groups) are more likely to be vaccinated if the number of cases increases. The impact is significant at the five per cent. This finding is almost in accordance with that found by Freeman, Loe [18], Kessels, Luyten [16] and Kourlaba, Kourkouni [28]. Respondents with comorbidities (X_21_) are less likely to be vaccinated than those who do not have any medical history. This behavior was observed in the South of Vietnam during the fourth wave [45]. The impact is at the one and five per cent level, depending on the pandemic severity. As previously addressed, the comorbidities and the new vaccines may make them hesitate to be vaccinated. This finding is almost similar to that discovered by Al-Mistarehi, Kheirallah [24], Dodd, Cvejic [23] and Yoda and Katsuyama [21], but different from that found by Abedin, Islam [22].

Respondents with sufficient knowledge about the virus fatality (X_26_) are more likely to be vaccinated when a new variant emerges. The impact is significant at the ten per cent level. There have been several variants of Coronavirus. It has been proven that they spread faster and are more lethal than the original, especially the Delta. Respondents who believe that herd immunity occurs if at least 70 per cent of the population is fully vaccinated (X_28_) are more likely to be vaccinated. The impact is significant at the one per cent level. Schwarzinger, Watson [19] examined this behavior, but the impact was not significant.

Respondents who have a family member or a friend or a co-worker infected with the virus (X_30_) are more likely to be vaccinated. The impact is significant at the one, five, and ten per cent level, depending on pandemic severity. Respondents who are afraid of being discriminated against if not vaccinated (X_31_) are more likely to be vaccinated than those who are not. The impact is significant at the one per cent level. Apart from volunteering to be vaccinated, laws and regulations (such as fines or imprisonment) applied to those who are not willing to be vaccinated have (in particular circumstances) gradually changed the vaccine behavior of the Vietnamese [48]. Respondents with the fear of catching the virus (X_32_) are less likely to be vaccinated if the pandemic becomes more severe. The impact is significant at the five per cent level. This is an unexpected result and future studies would be able to further examine this impact. Kelly and Southwell [20] showed that the impact of this factor was significant, but Khubchandani and Sharma [26] found it insignificant. Respondents with the fear of needles (X_33_) are more likely to be vaccinated. The impact is at the one per cent level if the pandemic becomes more severe and five per cent level if it becomes less severe. Perhaps the fear of catching the virus dominates that of needles. In addition, laser bubble guns with reasonable prices are expected to be available in the near future and should be able to combat this fear [49].

Despite being concerned about the information sufficiency (X_34_), side effects (X_35_) and safety of the vaccines (X_38_), respondents are still willing to be vaccinated against the virus. The impact is significant at the one and five per cent level, depending on the level of severity and type of concern. Al-Mistarehi and Kheirallah [24] found the impact of the concern about the information sufficiency of the vaccines was significant, but Kawata and Nakabayashi [31], Neumann-Böhme and Varghese [30] and Yoda and Katsuyama [21] found it to be insignificant. There are two possibilities to explain this behavior: the respondents had received sufficient information about the vaccines, or the fear of catching the virus dominates the concerns about insufficient information about the vaccines or the safety of the vaccines. Common reactions after being vaccinated are usually mild. However, severe reactions such as shocks have been observed and the consequences are worse [50,51]. Therefore, it is not surprising that there are several respondents who are concerned about the availability of the response procedure if there is a severe shock after being vaccinated. Sufficient information about the response procedure and the ways to minimize the risks should be available and accessible in multiple languages, including local languages. In addition, the procedure should be available in multiple formats such as video and audio. The regression results show that respondents who are concerned about the response procedure (X_36_) are less likely to be vaccinated. The impact is significant at the one and five per cent level, depending on the level of severity. Due to the fatality of the Delta variant, people rushed to be vaccinated while the vaccination sites were limited. In addition, due to the vaccine supply, the facilities’ readiness at the site and available nurses, the waiting time at the site may be long. As a result, vaccination sites are normally crowded and people have to wait. The longer the waiting time the higher the risk of catching the virus. Mobile vaccination should be able to cope with this challenge [46]. The results show that respondents who are concerned about the waiting time (X_41_) are less likely to be vaccinated. The impact is significant at the one per cent level. Al-Mistarehi and Kheirallah [24] also found similar behavior. An unsuitable time frame (X_42_) to vaccinate or an unsafe vaccination site or waiting room may discourage people from being willing to be vaccinated. The model of mobile vaccination should be sufficiently flexible to deal with this challenge [46]. Despite being concerned about the time frame inconvenience or safety of the vaccination waiting room to vaccinate, respondents are still more likely to be vaccinated if a new variant emerges. The impact is significant at the five per cent level. Knowing common symptoms after being vaccinated can help people prepare to deal with them, hence making them more confident in being vaccinated.

The results show that respondents with sufficient knowledge about common symptoms after getting vaccinated (X_44_) are more likely to be vaccinated if the number of deaths increases or a new virus variant emerges. The impact is significant at the ten and one per cent level, respectively.

The impact of nationality on the willingness to be vaccinated is significant at the one per cent level for all study countries. In addition, the odds ratio (OR) of respondents who are willing to be vaccinated if the pandemic becomes more severe slightly dominates that of those if it becomes less severe. Additionally, the OR of Indonesian (X_46_) and Malaysian (X_48_) respondents who are willing to be vaccinated significantly dominates that of the Vietnamese (X_45_) and Filipino (X_47_). The impact of the remaining variables on the willingness to be vaccinated is not statistically significant and can be further examined in future studies.

## 4. Conclusions

The current study uses data surveyed between August and September 2021 in four ASEAN countries to examine the willingness to be vaccinated against COVID-19 according to the pandemic severity. It also applies logistic regressions to examine the association between the willingness to be vaccinated with the influential factors.

The results show that the number of respondents who are willing to be vaccinated if the pandemic becomes more severe dominates that of those if it becomes less severe. In addition, the number of respondents with more work or family responsibilities who are willing to be vaccinated significantly dominates that of those without or with fewer work or family responsibilities. The number of respondents with protection such as health insurance or savings or assets who are willing to be vaccinated significantly dominates that of those without the protection. The number of respondents with better information access who are willing to be vaccinated significantly dominates that of those without or with poorer information access. The number of respondents who belong to the vaccine priority groups or are living with a person belonging to the vaccine priority groups significantly dominates that of those who do not belong to the groups or are not living with anyone belonging to the groups. In contrast, the number of respondents with comorbidities or the infection significantly dominates that of those without any comorbidity or infection. The number of respondents with the fear (of being infected with the virus, of being discriminated against if not vaccinated or of needles) who are willing to be vaccinated significantly dominates that of those without the fear. In contrast, the number of respondents who are not concerned about the vaccines and relevant matters and are willing to be vaccinated significantly dominates that of those who are.

Results generated from the logistic regressions show that the impact of selected influential factors on the willingness to be vaccinated with different levels of pandemic severity varies in terms of magnitude and direction. The impact of several factors on the willingness to be vaccinated is not significant and can be further examined. Due to resource limitations, the study cannot cover other countries or territories in ASEAN countries where the pandemic may also be severe.

## Figures and Tables

**Figure 1 vaccines-10-00222-f001:**
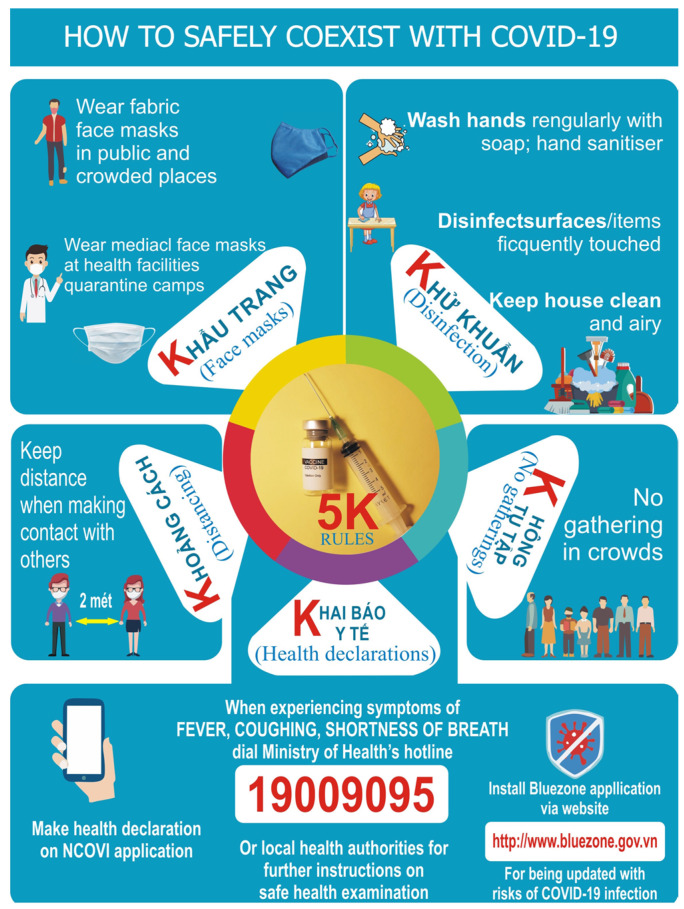
The 5K Message and Vaccination to Safely Co-exist with COVID-19 in Vietnam.

**Figure 2 vaccines-10-00222-f002:**
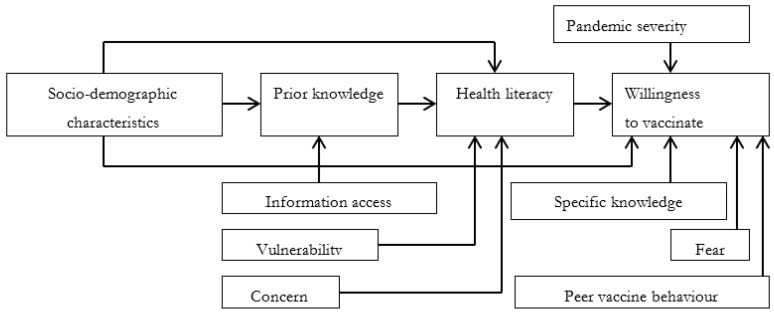
Conceptual framework on COVID-19 vaccine acceptance. Source. Illustrated by the authors with ideas adapted from Sun, Shi [37].

**Figure 3 vaccines-10-00222-f003:**
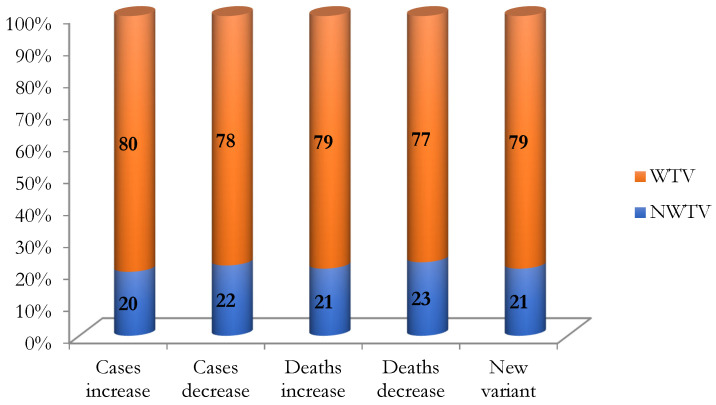
COVID-19 vaccine acceptance with different levels of pandemic severity. Source. Drawn by the authors using surveyed data. Note. WTV: Willing to be vaccinated, NWTV: Not willing to be vaccinated.

**Table 1 vaccines-10-00222-t001:** COVID-19 vaccine acceptance with different levels of pandemic severity.

Variable	Units/ Measures	Total	Cases Increase		Cases Decrease		Deaths Increase		Deaths Decrease		New Variant	
			NWTV ^1^	WTV ^2^	NWTV	WTV	NWTV	WTV	NWTV	WTV	NWTV	WTV
X_1_ ^3^	13–17	25	9	16	8	17	9	16	8	17	12	13
	18–24	658	132	526	154	504	141	517	156	502	141	517
	25–34	673	108	565	113	560	110	563	123	550	118	555
	35–44	726	136	590	142	584	149	577	142	584	141	585
	45–54	326	93	233	106	220	94	232	107	219	90	236
	55–64	66	9	57	10	56	9	57	10	56	9	57
	65–81	26	17	9	20	6	17	9	19	7	17	9
X_2_	0	1679	297	1382	334	1346	317	1363	345	1334	311	1368
	1	821	203	618	219	602	212	608	221	599	216	604
X_3_	0	1433	318	1115	354	1079	334	1100	357	1076	336	1097
	1	1067	182	885	198	869	195	872	210	857	192	875
X_4_	0	1007	237	770	268	739	253	754	266	741	254	753
	1	1493	263	1230	284	1209	276	1217	301	1192	274	1219
X_5_	0	1135	271	864	304	831	283	852	305	830	284	851
	1	1365	229	1136	249	1116	246	1119	262	1103	244	1122
X_6_	0	980	193	787	206	774	207	773	212	767	207	773
	1	1520	307	1213	347	1174	322	1199	354	1166	320	1200
X_7_	0	1786	348	1438	378	1408	374	1412	393	1393	370	1416
	1	714	152	561	175	539	155	559	173	541	158	556
X_8_	0	287	79	207	87	199	81	206	86	201	79	207
	1	2213	421	1793	465	1748	448	1765	481	1733	448	1765
X_9_	0	502	181	320	197	305	186	315	197	305	185	317
	1	1998	319	1679	356	1643	343	1656	370	1628	343	1656
X_10_	0	640	152	487	182	457	162	478	180	460	163	477
	1	1860	348	1513	370	1490	367	1493	387	1473	365	1496
X_11_	0	541	182	358	192	349	185	356	192	349	188	353
	1	1959	318	1641	361	1598	344	1615	375	1584	340	1619
X_12_	0	1499	330	1170	365	1135	347	1153	363	1136	348	1152
	1	1001	171	830	188	813	182	818	203	797	180	821
X_13_	< ^4^ 6	2148	470	1678	515	1634	495	1653	529	1619	490	1658
	≥ ^5^ 6	352	30	322	38	314	34	318	38	314	38	314
X_14_	<6	2208	476	1733	520	1688	500	1708	533	1675	495	1713
	≥6	292	25	267	33	259	29	263	34	258	33	259
X_15_	0	175	61	113	73	102	64	111	73	102	65	109
	1	2325	439	1886	479	1846	465	1860	494	1832	462	1863
X_16_	0	283	73	210	81	202	72	211	81	202	72	211
	1	2217	427	1790	472	1746	457	1760	486	1731	456	1761
X_17_	0	620	274	347	293	327	279	341	297	323	287	334
	1	1880	227	1653	259	1621	250	1630	270	1610	241	1639
X_18_	0	681	280	401	307	374	285	396	307	374	294	387
	1	1819	220	1598	245	1574	244	1575	259	1559	233	1585
X_19_	0	800	274	526	300	500	280	520	300	500	284	516
	1	1700	227	1473	253	1447	249	1451	267	1433	244	1456
X_20_	0	1279	339	941	370	909	347	933	386	894	350	929
	1	1221	162	1059	182	1038	182	1038	181	1040	177	1044
X_21_	0	1892	353	1539	392	1499	374	1518	410	1481	371	1520
	1	608	147	461	160	448	155	453	156	452	156	452
X_22_	0	752	197	555	210	542	198	554	214	538	198	554
	1	1748	304	1445	343	1406	331	1417	353	1395	330	1419
X_23_	0	421	113	307	111	310	108	313	115	306	106	315
	1	2079	387	1692	442	1638	421	1658	452	1627	422	1657
X_24_	0	1057	92	964	119	938	107	950	126	930	107	950
	1	1443	408	1036	434	1010	422	1021	440	1003	421	1023
X_25_	0	268	103	165	113	155	107	162	115	154	109	159
	1	2232	397	1834	439	1793	422	1810	452	1780	418	1813
X_26_	0	212	92	120	102	111	98	115	102	111	104	108
	1	2288	408	1880	451	1837	431	1856	465	1823	423	1864
X_27_	0	945	164	780	198	746	177	767	201	744	182	762
	1	1555	336	1219	354	1201	352	1204	366	1189	345	1210
X_28_	0	730	311	418	345	384	326	404	348	382	330	400
	1	1770	189	1582	207	1563	203	1567	219	1552	198	1572
X_29_	0	2316	469	1847	516	1800	496	1820	529	1787	496	1820
	1	184	31	152	36	147	33	151	38	146	31	152
X_30_	0	1688	374	1314	405	1283	401	1287	418	1270	397	1291
	1	812	126	685	147	664	128	684	149	663	130	681
X_31_	0	1308	387	921	416	892	403	905	427	881	396	912
	1	1192	113	1079	137	1055	126	1066	139	1053	132	1060
X_32_	0	1191	242	948	262	929	245	946	263	928	236	955
	1	1309	258	1051	291	1019	284	1025	304	1006	292	1017
X_33_	0	636	188	448	198	438	194	442	202	434	192	444
	1	1864	313	1552	354	1510	335	1529	365	1499	336	1528
X_34_	0	1704	276	1428	294	1410	298	1406	310	1394	296	1408
	1	796	224	572	258	538	231	565	257	539	232	564
X_35_	0	2209	293	1916	335	1875	317	1893	349	1860	313	1897
	1	291	207	83	218	73	212	78	218	73	215	76
X_36_	0	2208	300	1909	343	1866	324	1884	360	1849	327	1881
	1	292	201	91	210	82	205	87	207	85	201	91
X_37_	0	2224	304	1920	347	1877	327	1897	358	1866	324	1899
	1	276	197	79	206	70	202	74	208	68	203	73
X_38_	0	2224	296	1928	337	1886	317	1907	349	1875	318	1906
	1	276	205	72	215	61	212	64	218	59	210	66
X_39_	0	2243	306	1937	352	1892	327	1916	360	1884	327	1916
	1	257	194	63	201	56	202	55	207	50	201	56
X_40_	0	1979	288	1691	317	1662	310	1669	331	1648	306	1673
	1	521	212	309	236	285	219	302	236	285	221	300
X_41_	0	1903	285	1618	318	1585	306	1597	331	1572	307	1596
	1	597	215	382	234	362	223	374	236	361	220	376
X_42_	0	2021	283	1738	317	1704	306	1714	331	1690	302	1718
	1	479	218	262	236	244	223	257	236	244	225	254
X_43_	0	2035	287	1748	323	1712	310	1725	335	1700	310	1725
	1	465	214	251	229	236	219	246	232	233	218	248
X_44_	0	307	193	115	207	100	202	106	206	102	205	103
	1	2193	307	1885	345	1847	327	1866	361	1832	323	1869
X_45_	0	N/A ^6^	N/A	N/A	N/A	N/A	N/A	N/A	N/A	N/A	N/A	N/A
	1	651	218	433	225	426	220	431	225	426	220	431
X_46_	0	N/A	N/A	N/A	N/A	N/A	N/A	N/A	N/A	N/A	N/A	N/A
	1	667	54	613	68	599	65	602	75	592	65	602
X_47_	0	N/A	N/A	N/A	N/A	N/A	N/A	N/A	N/A	N/A	N/A	N/A
	1	651	74	578	89	562	75	576	89	562	74	578
X_48_	0	N/A	N/A	N/A	N/A	N/A	N/A	N/A	N/A	N/A	N/A	N/A
	1	530	267	263	278	252	274	257	279	251	274	257

*Source.* Calculated from the surveyed data. *Notes.* ^1^ Not willing to be vaccinated, ^2^ Willing to be vaccinated, ^3^ Please refer to the Abbreviations for details, ^4^ Fewer than and ^5^ Equal to or greater than and ^6^ Neither applicable nor available.

**Table 2 vaccines-10-00222-t002:** Association between COVID-19 vaccine acceptance and the influential factors.

Variable.	WTV ^1^ If Cases Increase		WTV If Cases Decrease		WTV If Deaths Increase		WTV If Deaths Decrease		WTV If New Variant	
	Sig. ^2^	OR ^3^	Sig.	OR	Sig.	OR	Sig.	OR	Sig.	OR
X_1_ ^4^ (ln ^5^)	0.003	0.089	0.000	0.017	0.001	0.071	0.000	0.041	0.021	0.163
95% C. I. ^6^	0.018	0.431	0.004	0.086	0.015	0.334	0.009	0.186	0.035	0.764
X_2_	0.605	0.907	0.756	0.945	0.831	0.962	0.787	0.953	0.659	0.923
95% C. I.	0.627	1.312	0.660	1.353	0.674	1.373	0.673	1.351	0.647	1.317
X_3_	0.039	1.445	0.005	1.621	0.044	1.419	0.038	1.412	0.006	1.617
95% C. I.	1.018	2.049	1.158	2.270	1.009	1.996	1.019	1.957	1.146	2.281
X_4_	0.651	0.896	0.362	1.243	0.713	1.091	0.817	0.948	0.711	0.916
95% C. I.	0.556	1.444	0.779	1.981	0.687	1.731	0.603	1.490	0.575	1.459
X_5_	0.471	1.166	0.377	1.206	0.589	1.119	0.429	1.176	0.481	1.159
95% C. I.	0.767	1.774	0.796	1.827	0.743	1.686	0.787	1.756	0.769	1.748
X_6_	0.479	0.877	0.307	0.831	0.649	0.922	0.175	0.789	0.781	0.951
95% C. I.	0.610	1.261	0.583	1.185	0.649	1.308	0.561	1.111	0.667	1.356
X_7_	0.337	0.807	0.036	0.641	0.314	0.804	0.220	0.777	0.203	0.758
95% C. I.	0.521	1.250	0.423	0.971	0.525	1.230	0.520	1.163	0.494	1.161
X_8_	0.468	1.038	0.106	1.084	0.335	1.049	0.080	1.089	0.173	1.071
95% C. I.	0.939	1.147	0.983	1.195	0.952	1.156	0.990	1.198	0.971	1.181
X_9_	0.281	1.248	0.120	1.369	0.166	1.324	0.146	1.333	0.113	1.380
95% C. I.	0.834	1.866	0.922	2.032	0.890	1.969	0.905	1.965	0.927	2.056
X_10_	0.351	0.797	0.161	1.396	0.836	0.952	0.309	1.267	0.601	0.883
95% C. I.	0.494	1.284	0.876	2.225	0.596	1.519	0.803	2.000	0.553	1.409
X_11_	0.107	0.684	0.009	0.539	0.013	0.559	0.016	0.571	0.025	0.590
95% C. I.	0.431	1.086	0.339	0.856	0.352	0.886	0.362	0.901	0.372	0.937
X_12_	0.800	0.953	0.996	1.001	0.938	1.014	0.297	0.831	0.805	1.047
95% C. I.	0.653	1.388	0.697	1.438	0.708	1.454	0.586	1.177	0.729	1.503
X_13_	0.598	0.954	0.843	1.017	0.826	0.981	0.542	1.052	0.584	0.955
95% C. I.	0.799	1.138	0.862	1.200	0.828	1.163	0.894	1.238	0.810	1.126
X_14_	0.027	1.246	0.248	1.111	0.066	1.190	0.608	1.047	0.104	1.160
95% C. I.	1.025	1.514	0.929	1.328	0.989	1.433	0.879	1.246	0.970	1.386
X_15_	0.769	0.886	0.060	2.113	0.398	1.420	0.057	2.093	0.122	1.917
95% C. I.	0.396	1.984	0.970	4.605	0.630	3.202	0.977	4.482	0.841	4.370
X_16_	0.230	1.515	0.620	0.840	0.996	0.998	0.712	0.881	0.445	0.755
95% C. I.	0.769	2.983	0.423	1.670	0.494	2.017	0.451	1.723	0.367	1.552
X_17_	0.826	1.092	0.177	0.609	0.740	0.875	0.684	0.863	0.789	0.900
95% C. I.	0.500	2.384	0.296	1.251	0.398	1.923	0.425	1.753	0.418	1.939
X_18_	0.380	1.402	0.005	2.688	0.330	1.461	0.056	1.938	0.116	1.810
95% C. I.	0.659	2.984	1.357	5.327	0.682	3.128	0.982	3.824	0.864	3.792
X_19_	0.280	1.232	0.035	1.481	0.287	1.224	0.102	1.345	0.110	1.354
95% C. I.	0.844	1.799	1.028	2.135	0.843	1.778	0.943	1.920	0.934	1.963
X_20_	0.698	1.070	0.706	1.066	0.386	0.863	0.150	1.263	0.826	0.964
95% C. I.	0.759	1.509	0.766	1.483	0.619	1.203	0.919	1.736	0.692	1.342
X_21_	0.022	0.654	0.022	0.661	0.019	0.656	0.153	0.777	0.005	0.601
95% C. I.	0.455	0.939	0.464	0.942	0.461	0.933	0.550	1.098	0.423	0.855
X_22_	0.714	1.078	0.948	0.987	0.541	1.130	0.897	0.975	0.940	0.985
95% C. I.	0.722	1.610	0.669	1.457	0.764	1.673	0.669	1.421	0.667	1.454
X_23_	0.989	0.997	0.129	1.419	0.229	1.316	0.295	1.261	0.110	1.446
95% C. I.	0.639	1.555	0.903	2.231	0.841	2.058	0.817	1.945	0.920	2.274
X_24_	0.223	1.301	0.676	1.091	0.154	1.340	0.494	1.146	0.137	1.359
95% C. I.	0.852	1.987	0.725	1.642	0.896	2.003	0.775	1.696	0.907	2.036
X_25_	0.481	1.235	0.475	1.226	0.569	1.183	0.319	1.324	0.998	1.001
95% C. I.	0.687	2.221	0.701	2.145	0.663	2.114	0.763	2.299	0.560	1.789
X_26_	0.903	0.960	0.491	1.245	0.672	1.148	0.749	1.107	0.100	1.689
95% C. I.	0.500	1.844	0.668	2.323	0.605	2.180	0.594	2.064	0.904	3.156
X_27_	0.321	1.200	0.770	0.950	0.312	1.195	0.722	1.062	0.480	1.133
95% C. I.	0.837	1.719	0.675	1.338	0.846	1.689	0.763	1.479	0.802	1.600
X_28_	0.001	1.917	0.000	2.558	0.000	2.090	0.000	2.354	0.000	2.247
95% C. I.	1.309	2.806	1.787	3.662	1.451	3.011	1.661	3.335	1.560	3.238
X_29_	0.483	1.272	0.563	1.209	0.699	1.141	0.778	1.094	0.356	1.377
95% C. I.	0.650	2.490	0.635	2.300	0.585	2.224	0.588	2.036	0.698	2.715
X_30_	0.024	1.771	0.100	1.504	0.003	2.085	0.070	1.560	0.015	1.851
95% C. I.	1.078	2.910	0.925	2.446	1.275	3.409	0.965	2.522	1.127	3.041
X_31_	0.000	2.784	0.000	2.671	0.000	2.522	0.000	2.535	0.000	2.221
95% C. I.	2.149	3.605	2.086	3.419	1.968	3.232	1.995	3.221	1.738	2.839
X_32_	0.042	0.705	0.267	0.832	0.337	0.854	0.360	0.864	0.916	1.018
95% C. I.	0.504	0.987	0.600	1.152	0.618	1.179	0.631	1.182	0.736	1.407
X_33_	0.006	1.683	0.049	1.442	0.004	1.699	0.034	1.468	0.040	1.466
95% C. I.	1.162	2.440	1.001	2.077	1.183	2.439	1.029	2.095	1.018	2.110
X_34_	0.001	1.777	0.000	2.436	0.004	1.618	0.000	2.062	0.002	1.672
95% C. I.	1.270	2.486	1.758	3.375	1.168	2.241	1.506	2.824	1.206	2.316
X_35_	0.471	1.481	0.387	1.553	0.546	1.390	0.422	1.530	0.006	4.344
95% C. I.	0.509	4.313	0.573	4.208	0.478	4.043	0.542	4.315	1.524	12.381
X_36_	0.913	1.068	0.909	0.936	0.296	0.515	0.022	0.228	0.001	0.097
95% C. I.	0.328	3.474	0.297	2.943	0.148	1.788	0.064	0.810	0.024	0.403
X_37_	0.901	0.924	0.609	0.734	0.641	0.742	0.923	1.062	0.338	1.812
95% C. I.	0.269	3.173	0.224	2.401	0.212	2.602	0.316	3.571	0.538	6.108
X_38_	0.002	6.274	0.000	8.191	0.000	8.331	0.000	7.625	0.034	3.483
95% C. I.	1.969	19.990	2.738	24.508	2.762	25.125	2.541	22.878	1.099	11.042
X_39_	0.338	1.796	0.624	0.736	0.056	3.235	0.026	3.818	0.075	3.120
95% C. I.	0.542	5.943	0.215	2.511	0.968	10.809	1.176	12.401	0.890	10.931
X_40_	0.400	0.727	0.587	1.191	0.363	0.711	0.921	0.968	0.756	0.892
95% C. I.	0.346	1.527	0.634	2.240	0.341	1.483	0.509	1.840	0.435	1.832
X_41_	0.003	0.339	0.010	0.440	0.006	0.384	0.010	0.442	0.001	0.293
95% C. I.	0.167	0.686	0.235	0.823	0.194	0.759	0.238	0.823	0.146	0.587
X_42_	0.255	1.741	0.154	1.891	0.511	1.379	0.450	1.406	0.049	2.565
95% C. I.	0.670	4.524	0.787	4.541	0.529	3.594	0.581	3.401	1.005	6.548
X_43_	0.628	1.293	0.432	0.684	0.882	0.924	0.767	1.153	0.032	0.305
95% C. I.	0.458	3.651	0.265	1.765	0.325	2.629	0.451	2.951	0.103	0.903
X_44_	0.864	1.097	0.066	2.353	0.137	2.138	0.535	1.350	0.001	5.335
95% C. I.	0.383	3.142	0.945	5.855	0.785	5.824	0.524	3.479	2.021	14.080
X_45_	0.760	0.932	0.764	0.934	0.868	0.963	0.697	1.090	0.821	0.950
95% C. I.	0.592	1.467	0.597	1.460	0.619	1.499	0.706	1.684	0.612	1.477
X_46_	0.000	9.502	0.000	6.751	0.000	8.811	0.000	6.287	0.000	7.191
95% C. I.	5.050	17.877	3.680	12.386	4.785	16.225	3.475	11.372	3.914	13.214
X_47_	0.029	2.031	0.011	2.240	0.003	2.573	0.018	2.081	0.005	2.474
95% C. I.	1.076	3.834	1.201	4.178	1.373	4.820	1.132	3.827	1.316	4.648
X_48_	0.000	6.915	0.000	4.510	0.000	6.939	0.000	4.564	0.000	6.811
95% C. I.	3.588	13.324	2.417	8.413	3.682	13.074	2.480	8.401	3.566	13.010
Constant	0.458	0.421	0.839	1.265	0.351	0.343	0.510	0.480	0.041	0.094

*Source.* Calculated from the surveyed data. *Note.* ^1^ Willingness to be vaccinated, ^2^ Significance, ^3^ Odds ratio, ^4^ Please refer to the Abbreviations for details, ^5^ Natural log, ^6^ Confidence interval.

## Data Availability

The datasets generated and/or analyzed during the current study are not publicly available due to the sensitive nature of the questions asked (especially those on demographic characteristics and personal perceptions), but are available from the corresponding author on reasonable request.

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
