# Peer review of "COVID-19 Vaccine Acceptance among ASEAN Countries: Does the Pandemic Severity Really Matter?"

_vaccines, 2022, doi:10.3390/vaccines10020222_

Round 1

Reviewer 1 Report

This study aimed to identify the driver factors for acceptance of COVID-19 vaccination in the Association of Southeast Asian Nations (ASEAN) under different levels of pandemic severity, using data from a questionnaire-based survey conducted in August and September 2021 in four countries: Indonesia, Malaysia, the Philippines, and Vietnam. However, although several influencing factors have been identified, logistic analysis did not show significant differences in many of them, and no specific recommendations have been made. From the content of the questions, it seems that there is not much potential for future development as research.

Comments

According to this study, 78-80% of the target population is vaccination acceptance, regardless of the severity of the pandemic. Nonetheless, the vaccination rate at the end of April is very low. Why is this? Isn't it a matter of people's will that vaccination rates are not increasing?

The title of this study is "COVID-19 vaccine acceptance in ASEAN," but only four countries (Indonesia, the Philippines, Malaysia, and Vietnam) are actually involved. These countries have a high number of COVID-19 cases and deaths, and low vaccination rates. Can the bias of deliberately selecting such countries be considered negligible?

Also, the above four countries do not necessarily represent ASEAN. Therefore, it seems inappropriate to use "ASEAN" in the title.

The results are just listed in the text, and it is difficult to grasp the content. A table summarizing the results should be prepared.

It is difficult to understand which part of the Table contains the results described in the text. Please show the rows of the Table that should be looked at in the text.

In the text, it says that the vaccines were administrated to people aged 15-64 years, so people in that age range were sampled, but in Table 1, the age range is 13-81 years. Please explain.

The total number of people in each age group in row X1 of table 1 is 2501, which is different from the 2500 in other sections. Please explain.

Author Response

Dear Reviewer 1,

Thank you very much for your comments and questions. Please find our responses and answers in the file attached.

Best wishes,

Duong Hoai An

Reviewer 2 Report

The manuscript by An et al., is a surveyed study conducted the past summer in Vietnam, Indonesia, the Philippines and Malaysia concerning the COVID-19 vaccine acceptance according to the pandemic severity between the population. The authors also analyzed several influential factors to examine the willingness to vaccinate against the Coronavirus.

Given the complexity involved, the author has produced many positive and welcome outcomes. Overall, this research is well written and is well-balanced. The content of this manuscript is of major interest. Nevertheless, the following issues need to be addressed.

I do not find any significant incorrectness. My following comments are of minor character.

Introduction: You should change the second “On the other hand” with a synonymous

Please check the whole text for typos, such as spaces, absence of spaces between the word, etc…

In my opinion, the reference list style is wrong. Please check and correct it

Author Response

Dear Reviewer 2,

A: Dear Reviewer 2. Thank you very much for your comments. Thanks to your comments we have a great chance to improve our manuscript. We  have followed your comments to edit it as follows.
Q: The manuscript by An et al., is a surveyed study conducted the past 
summer in Vietnam, Indonesia, the Philippines and Malaysia concerning the 
COVID-19 vaccine acceptance according to the pandemic severity between 
the population. The authors also analyzed several influential factors to 
examine the willingness to vaccinate against the Coronavirus.

Given the complexity involved, the author has produced many positive and 
welcome outcomes. Overall, this research is well written and is wellbalanced. The content of this manuscript is of major interest. Nevertheless, 
the following issues need to be addressed.
I do not find any significant incorrectness. My following comments are of 
minor character.
Q: Introduction: You should change the second “On the other hand” with a 
synonymous
A: Thank you very much for your comment. The two sentences have been 
edited as: “The hesitation and behaviour challenge us to reach the herd 
immunity. In addition, this behaviour creates a chance for more variants 
such as Delta or Omicron to develop and will be more difficult to deal 
with.”
Q: Please check the whole text for typos, such as spaces, absence of spaces 
between the word, etc…
A: Thank you very much for your comment. We have tried our best to
manually checked and corrected the typos as many as we can. The “ 
Grammarly” has been used to check and correct any typos left.
Q: In my opinion, the reference list style is wrong. Please check and correct 
it
A: Thank you very much for your comment. We have tried our best to 
check and fix the errors to make sure it follows the guides of the Journal
(https://mdpi-res.com/data/mdpi_references_guide_v5.pdf). In addition, 
the Digital Objective Identifiers (DOI) of the cited articles (in the 
References) have been added as guided.

Best wishes,

Duong Hoai An

Reviewer 3 Report

Dear Authors: thank you for the opportunity to read your paper. You address an important and a very relevant topic. My suggestions, beneath, are meant to make your work more impactful. This is what I suggest you should do:

--the Introduction: at present it is very short and does not offer the reader an insight into the problem at hand. Why would our readers think that vaccination is an issue? From a different angle, why would readers engaged with geographical areas other than ASEAN be interested in your paper? Please, make sure that in the Introduction to outline the diverse aspects and variability of citizens' willingness (or not) to get vaccinated. If you do that, then you will be able to argue that the findings of your research are relevant and transferrable to other parts of the world. 

--What follows is that if you restructure and expand the Introduction, also your conclusion will take a clearer shape and direction. That is, finally you will be able to address the question 'so what'. Please, consider. 

-- As regards technical issues: (i) acronyms, e.g. ASEAN: first the full name, then the acronym; the only exception will be the title; (b) make sure that you divide your discussion into paragraphs so that it is easier to read; (c) in your discussion on the 4 ASEAN countries that you selected: the case of Vietnam is elaborated quite extensively, while the remaining three cases are not. I think it would be important to mention that in Malaysia for instance the lockdown lasted considerably longer than in other countries, etc. (d) You may want to reflect on the already existing literature on Covid, vaccination, ASEAN etc., e.g.  https://www.emerald.com/insight/content/doi/10.1108/TG-08-2020-0193/full/html  

Author Response

Dear Reviewer 3, 

Thank you very much for your comments. Thanks to your comments we have a great chance to improve our manuscript. We have followed your comments and tried our best to edit it as follows.
Q: Dear Authors: thank you for the opportunity to read your paper. You address an important and a very relevant topic. My suggestions, beneath, are meant to make your work more impactful. This is what I suggest you should do:
Q: --the Introduction: at present it is very short and does not offer the reader an insight into the problem at hand. Why would our readers think that vaccination is an issue? From a different angle, why would readers engaged with geographical areas other than ASEAN be interested in your paper? Please, make sure that in the Introduction to outline the diverse aspects and variability of citizens' willingness (or not) to get vaccinated. If you do that, then you will be able to argue that the findings of your research are relevant and transferrable to other parts of the world.
A: Thank you very much for your comment. As your comments, the Introduction has been expanded. Particularly, it has addressed the importance of the COVID-19 vaccines, the distinction of the study locations and clearly stated the research questions, which will be answered.
Q: --What follows is that if you restructure and expand the Introduction, also your conclusion will take a clearer shape and direction. That is, finally you will be able to address the question 'so what'. Please, consider.
A: Thank you very much for your comment.
Q: -- As regards technical issues:
(i) acronyms, e.g. ASEAN: first the full name, then the acronym; the only exception will be the title
A: Thank you very much for your comment. The manuscript has been check to follow this rule.
(b) make sure that you divide your discussion into paragraphs so that it is
easier to read;
A: Thank you very much for your comment. The discussion has been split into paragraphs where applicable.
(c) in your discussion on the 4 ASEAN countries that you selected: the case of Vietnam is elaborated quite extensively, while the remaining three cases  are not. I think it would be important to mention that in Malaysia for instance the lockdown lasted considerably longer than in other countries, etc
A: Thank you very much for your comment. An article has been cited and a sentence has been added to elaborate the case of Malaysia.
(d) You may want to reflect on the already existing literature on Covid, vaccination, ASEAN etc.,
e.g. https://www.emerald.com/insight/content/doi/10.1108/TG-08-2020-
0193/full/html
A: This article and another have been cited to make the case of Indonesia
worth to examine.

Best wishes,

Duong Hoai An

Round 2

Reviewer 1 Report

The manuscript has been improved through an exchange of comments and responses. 

I would add one thing.

The reason for the change in the age range of the survey is stated on P6, L203-205, but I think it would be more appropriate to explain it properly on P3. 2.2.2.  sampling section.

Author Response

Dear Reviewer 1,

Thank you very much for giving us another chance to improve our manuscript. The manuscript has been revised based on your comments.

Dear Reviewer 1, we realised the issue right after submitting the first
revision. However, we thought we would be given another chance to fix it.
Thank you very much for giving us that change. Particularly, the paragraph
that explains some age outliers has been moved to the Sampling section
(p.3, 2.1.2).

Best wishes,

An

Reviewer 3 Report

Thank you for addressing some of my comments.

Author Response

Dear Reviewer 3,

Thank you very much.

Best wishes,

Duong Hoai An
